# Years of Potential Life Lost on Renal Replacement Therapy: Retrospective Study Cohort

**DOI:** 10.3390/jcm12010051

**Published:** 2022-12-21

**Authors:** José Manuel Muñoz-Terol, José L. Rocha, Pablo Castro-de la Nuez, Emilio García-Cabrera, Ángel Vilches-Arenas

**Affiliations:** 1Department of Nephrology, Hospital Universitario Virgen del Rocío, 41013 Seville, Spain; 2Department of Medicine, University of Seville, 41009 Seville, Spain; 3Information System of the Autonomic Transplant Coordination of Andalusia (SICATA), 41004 Seville, Spain; 4Preventive Medicine and Public Health Department, University of Seville, 41009 Seville, Spain; 5Department of Preventive Medicine, Hospital Universitario Virgen Macarena, 41009 Seville, Spain

**Keywords:** renal replacement therapy, end-stage kidney disease, kidney transplantation, premature mortality

## Abstract

Background: Chronic kidney disease is the non-communicable disease with the highest growth in morbidity and mortality. Renal transplantation (RT) is the first option of renal replacement in end-stage kidney disease (ESKD) and dialysis is an alternative. However, there is no objective quantification of the impact of both options on a patient’s overall survival. The purpose of our study is to assess the potential years of life lost by patients on renal replacement therapy. Methods: Retrospective study cohort conducted from 2008 to 2018 based on autonomic data registry. Results: 11,551 patients included who received renal replacement therapy (RRT) in a range of age from 15 to 94 years. The mean age at the time of onset was 62.7 years, 95% confidence interval (95% CI) (62.4; 63.0). The mortality rate of RRT patients was 42.2%, 95% CI (41.5; 43.3) and the mean age at death was 72.7 years, 95% CI (72.4; 73.1). The number of patients with ESKD treated with RT was 3776, 32.7% of the total, 95% CI (31.8; 33.5). The total amount of years of potential life lost (YPLL) in the entire cohort was 77,831.3 years, 48,010.1 years in men, and 29,821.2 years in women. The mean number of YPLL per patient with RRT was 6.74 years in both sexes, 6.95 years in women, and 6.61 years in men. The mean number of potential years of life lost in dialysis patients was 9.0 years in both sexes, 8.8 years in men, and 9.2 years in women, while among kidney transplant recipients this figure decreased to 2.2 years in both men and women. Conclusions: End-stage chronic kidney disease in renal replacement therapy by dialysis causes an average of 9.0 years of life potentially lost for each patient on dialysis treatment, while having received a kidney transplant reduces this figure by 75.6%.

## 1. Introduction

During the last three decades, chronic kidney disease (CKD) has become the fastest growing noncommunicable disease in the world. According to the latest report of the Global Burden of Disease Study (GBDS) in 2017, between 649.2 and 752.1 million people in the world live with CKD at any stage [1].

Renal replacement therapy (RRT), whether hemodialysis (HD), peritoneal dialysis (PD), or renal transplantation (RT), is life-sustaining for patients with end-stage chronic kidney disease (ESKD). A study by Liyanage and Ninomiya [2] estimated the global prevalence of RRT in 2010, with 2.6 million people worldwide receiving some form of RRT. Furthermore, dialysis has increased globally by 43.1% between 1990 and 2017 and kidney transplantation by 34.4% [1].

Renal transplantation is currently the best RRT option that can be applied to patients with advanced ESKD, due to lower mortality [3,4] and better patient survival [5,6,7,8,9,10,11] as well as better cost-effectiveness [12,13]. Access to kidney transplantation is limited primarily by organ availability. According to the latest recommendations [14], the same clinical criteria (ESKD), age, and comorbidity are the determining factors in the choice of organ recipient. For this reason, an increasing number of renal transplants are performed each year on elderly patients [15,16]. However, there is no fixed age pattern nor objective indicators to measure survival loss between the two RRT options at different age of onset.

Years of potential life lost (YPLL) is becoming a valuable measure for public health surveillance and also a very useful way to measure mortality from noncommunicable diseases, especially for assessing premature mortality. It is also a measure that can allow comparison of mortality outcomes for a pathology or risk factor between different places (provinces, regions, countries, or regions of the world) and also enables comparison of outcomes between different pathologies [17].

The purpose of our study is to assess the potential years of life lost by patients on renal replacement therapy.

## 2. Materials and Methods

We conducted a retrospective cohort study from 1 January 2008 to 31 December 2019 in all centers located in Andalusia, southern Spain, covering a population of 8.4 million inhabitants. We included all incident patients who started RRT during the study period with at least one year of follow-up. We excluded pediatric patients, patients with cardiorenal syndrome, patients with RRT at another stage different from ESKD, and patients who started RRT outside of Andalusia.

The data source for RRT patients was the basic CKD module of SICATA [18]. SICATA is a mandatory population-based registry, which is attached to the Andalusian Autonomous Transplant Coordination Office. Patients are discharged upon receiving their first RRT, whether it is hemodialysis (HD), peritoneal dialysis (PD), or renal transplantation (RT).

All patients, when starting RRT in the different hospitals and dialysis centers in Andalusia, signed an informed consent document that informed them of the inclusion of their data in SICATA and the possible conducting of studies for epidemiological and research purposes. The study was carried out according to ethical principles and according to the requirements expressed in the Declaration of Helsinki (Fortaleza-Brazil Review, October 2013) [19] approved by Virgen del Rocio-Macarena Hospitals Ethical Committee, date of approval 26 January 2021.

All records contain the following variables: sex, age at the time of onset of RTT, adjusted Charlson index KD [20], labor status, and mortality data. 

YPLLs were obtained, first, by calculating the number of patients in RRT who died from any cause by age and sex in Andalusia for each study year (2008–2018). Second, the life expectancy in Andalusia was extracted at the age of the onset of RRT for men and women in the study period, according to data available from the Institute of Statistics and Cartography of Andalusia (IECA) [21]. Third, for each death case, the actual age of death of each patient was subtracted from the life expectancy, differentiating the results by sex. Subsequently, the number of patients who died in each year and at each age was multiplied by the number of years lost in each case, and, in this way, the total number of potentially lost years of life for each age and sex was obtained.

Finally, the results were grouped by age at onset, in groups of 5 years, except in patients between 15 and 29 years of age who were unified due to the few cases of death in this population segment, and divided by the number of patients with RRT in each age grouping, thus calculating the rate of years of life potentially lost for each patient with RRT. We segmented the results according to whether they received dialysis treatment (DT) or whether they received RT in their history of RRT and men and women were always considered separately.

Comparisons between patients who started dialysis and hemodialysis were made for qualitative variables using the Chi-square test, and for quantitative variables, the parametric t-test was used. For the calculation of population (95% CI), the normal approximation was used. All calculations were performed using the statistical software R version (4.03) [22].

## 3. Results

During the study period, a total of 11,551 patients received RRT in a range of age from 15 to 94 years. The disease that caused the RRT is listed in the Appendix A. The mean age at the time of onset was 62.7 years, 95% CI (62.4; 63.0). The predominant sex was male, 62.8%, 95% CI (61.9; 63.7), and 20.9% of them were active at work, 95% CI (19.8; 21.9). The mean of the comorbidity index was 6 points, 95% CI (5.9; 6) ranging from 2 to 20. The mortality rate of RRT patients was 42.2%, 95% CI (41.5; 43.3) and the mean age at death was 72.7 years, 95% CI (72.4; 73.1). The causes of death are detailed in the Appendix A.

### 3.1. Kidney Transplantation

A total of 3776 patients, 32.7% of the entire ESKD patient studied cohort, 95% CI (31.8; 33.5), were receiving RT. Comparing patients who received RT versus DT, there was a mean of 17.3 years younger, 95% CI (16.8; 17.8), a mean of 3.2 points less in the comorbidity index, 95% CI (3.1; 3.3), a 49% lower mortality rate, 95% CI (47%; 50%), and younger age at death by 7.9 years, 95% CI (7.5; 8.3) (Table 1).

### 3.2. Years of Potential Life Lost

The total amount of YPLL in the entire cohort was 77,831.3 years, 48,010.1 years in men and 29,821.2 years in women. Over time, there was a reduction in YPLL between 2008 and 2018 of 4669.7 years in men and 3343.7 years in women. The maximum impact of potential years of life lost in men occurred among incident RRT patients between 55 and 79 years of age and in women between 60 and 79 years of age. (Figure 1). The complete data table is included in the Appendix A.

The mean number of YPLL per patient with RRT was 6.74 years in both sexes, 6.95 years in women, and 6.61 years in men. The highest rate occurred between 50 and 69 years of age in men and between 55 and 74 years of age in women. The maximum value of the YPLL rate per patient in men was 8.14 years/patient with age at the beginning of the RRT between 65 and 69 years, and in women it was 8.85 years/patient with age at the beginning of the RRT between 60 and 64 years (Table 2).

### 3.3. YPLL Kidney Transplantation Versus Dialysis

The mean number of potential years of life lost in dialysis patients was 9.0 years in both sexes, 8.8 years in men, and 9.2 years in women, while among kidney transplant recipients, this figure decreased to 2.2 years in both men and women (Table 3). Analysis by age group shows that dialysis patients have a higher early mortality between the ages of 30 and 55 years while RT patients have the highest impact in the age group of 60 to 69 years (Table 3).

## 4. Discussion

Assessment of mortality by potential years of life lost, a measure that the Global Burden Disease Group (GBDG) has used extensively in its reports [5,23], has been little used in the CKD setting. The information provided by YPLL can be considered complementary to mortality and survival, offers a public health view of the CKD problem, and could allow comparison of results between different populations and of the impact on mortality between different pathologies [17].

In our study, the absolute number of YPLL per incident patient in RRT was 77,831.3 years, with a higher number of YPLL in men (61.7% of the total). However, when the rate of YPLL per patient in RRT was calculated, it was observed that the impact of RRT in ESKD patients was greater in women than in men, with 6.95 YPLL versus 6.61 YPLL, respectively. These data are in line with the latest report published by the United States Renal Data System [6]. Females receiving HD treatment do not have a significant survival advantage compared to the general population [24]. The prognostic factors associated with the female gender that cause this decrease in survival remain controversial. A less protective effect of body mass index (BMI) on HD mortality, associated with a different rate of skeletal muscle and fat due to sex hormones, is one of the most recent hypotheses [25]. However, the effect of diabetes is the most reported interaction factor on the different survival rates observed between sexes in all causes of death [24,25,26].

YPLLs for each patient who died in RRT were 15.9 years for both sexes, 15.3 in men, and 16.9 in women. The magnitude of premature death in RRT has a similar impact to that of COVID-19, measured by mortality in 81 countries, with a mean of 16 years per death, higher in the elderly [27].

YPLLs accumulate at the age of incidence between 55 and 79 years in men and between 60 and 79 years in women. Every woman aged 60–69 who initiates RRT loses 8.8 potential years of life, and every incident RRT man aged 50–69 loses 8 potential years of life. Furthermore, patients who started RRT with hemodialysis lost, on average, 8.8 years in men and 9.2 in women, but kidney transplant recipients’ life only reduced by 2.2 years, on average. These data are consistent with a recent Italian cohort over the same study period, but without sex stratification [28]. They reported a 5% annual increase in APVP from 40 to 70 years of age in HD patients, while RT patients remained stable.

Our data show that in the elderly population, (between 65 and 79 years of age) renal transplantation reduces the potential years of life lost by half compared to dialysis. The benefit of renal transplantation in the elderly had already been reported in older studies, such as that performed by Wolfe with data from the United States Renal Data System (USRDS) in incident patients between 1991 and 1996 [29]. A step further is the study by Cabrera and collaborators, from Spain, which was carried out with 138 patients over 75 years of age and with various comorbidities to whom organs were transplanted from donors with an average age of 77 years, resulting in a patient survival rate of 82, 1% at 1 year and 60.1% at 5 years, suggesting that renal transplantation should be considered in patients in need of RRT over 75 years of age and without absolute contraindications for renal transplantation [30]. Transplantation in older people is nowadays already a real trend, as the performance of transplantation in patients older than 65 years has grown from 7 to 22% [31], with a clinical outcome similar to that of dialysis patients [15]. Conservative treatment agreed with the family and the patient [3] while on the waiting list could be the best option, if renal function is clinically stable.

Our study has some limitations; the first is that all included patients had, at the time of inclusion, at the discretion of their responsible physicians, an indication for uninterrupted initiation of RRT, but these criteria may not have been completely uniform between the different treatment centers and, moreover, during the time of the study may have varied. However, the study period was recent and not long enough in duration as to be influenced by significant technical advances in the types of RRT used, and it was carried out in a homogeneous geographical area, Andalusia, with a large population base of more than 8 million people. Second, the type of data analyzed are data from an observational patient registry; nevertheless, these data were collected by the doctors responsible for the patients in each treatment center and the inclusion of patients in the register is mandatory in order to authorize payments for their treatment, which ensures that the inclusion of cases is complete and that there is no significant loss of incident cases in RRT. Third, there were only two patients over 80 years of age in our cohort receiving renal transplantation, so it is possible that the real difference between HD and RT is overestimated; however, recent published data [32] show better survival rates over RT patients, in line with our results. Finally, estimated years of life potentially lost have been fully attributed to RRT for all causes of death, therefore, no competing risks have been considered and a likely overestimation of YPPLL due to CKD in RRT has been assumed. This limitation did not confer bias in the comparison between hemodialysis and kidney transplantation because the overestimation of mortality is included in both groups and the entire population of our study is end-stage patient kidney disease (ESKD) treated with RRT. This higher mortality risk of patients with RRT can be considered to be associated with CKD, but it is indistinguishable from that which would occur in this population if they did not require RRT, so this division, in this case, is impossible to carry out exhaustively. Other series use the same criteria [3,6,28,29,30], and this is why the mortality rate did not differ and is comparable.

## 5. Conclusions

In conclusion, end-stage chronic kidney disease in dialysis-induced renal replacement therapy causes an average of 9.0 years of life potentially lost for each patient on dialysis treatment, while among kidney transplant recipients, this figure decreased to 2.2 years. Since the limiting factor of organ availability cannot be ignored, greater monitoring or specific programs for elderly people, especially diabetic women, must be implemented to mitigate the impact of premature death due to dialysis.

## Figures and Tables

**Figure 1 jcm-12-00051-f001:**
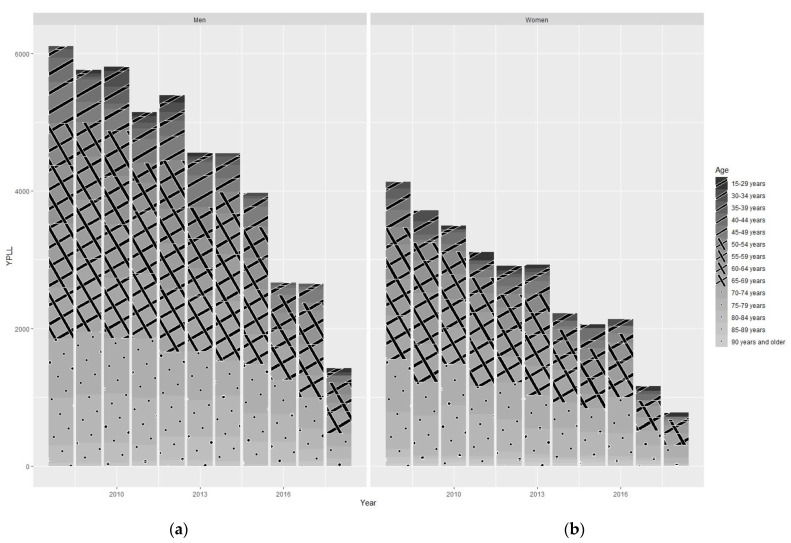
Men’s and women’s distribution of years of potential life lost (YPLL) during the study period: (**a**) men; (**b**) women. Age distribution: 15 to 49, one line; 50 to 69, crossed lines; 70 to 90, points.

**Table 1 jcm-12-00051-t001:** Demographic characteristics of patients included.

N = 11,551	Renal Transplantation N = 3776			No Renal Transplantation N = 7775			*p* Value
	Min–Max	Mean (SD)	95% CI	Min–Max	Mean (SD)	95% CI	
**Age (years)**	**15–80**	**51 (13.5)**	**(50.6; 51.5)**	**15–94**	**68.3 (12.6)**	**(68.0; 68.6)**	**<0.001**
**Sex**							
Women N (%)		**1371 (36.3)**	**(34.8; 37.9)**		**2922 (37.6)**	**(36.5; 38.7)**	**0.184**
Men N (%)		**2405 (63.7)**	**(62.1; 65.2)**		**4853 (62.4)**	**(61.3; 63.5)**	
**Charlson index**	**2–13**	**3.8 (1.8)**	**(3.8; 3.9)**	**2–20**	**7.0 (2.4)**	**(7.0; 7.1)**	**<0.001**
**Vital status**							
Alive N (%)		**3413 (90.4)**	**(89.4; 91.3)**		**3238 (41.6)**	**(40.6; 42.7)**	**<0.001**
Dead N (%)		**363 (9.6)**	**(8.7; 10.6)**		**4537 (58.4)**	**(57.3; 59.4)**	
**Age at death * (Years)**	**15–84**	**65.4 (10.2)**	**(64.3; 66.5)**	**19–98**	**73.3 (11.1)**	**(73.0; 73.6)**	**<0.001**

* Age at the time patient died.

**Table 2 jcm-12-00051-t002:** Years of potential life lost (YPLL) in men and women according to age of onset of renal replacement therapy.

	MEN			WOMEN		
Age Onset RRT	Total YPLL	N Patient	YPLL/ Patient	Total YPLL	N Patient	YPLL/ Patient
15–29 years	390.3	218	**1.79**	486.6	166	**2.93**
30–34 years	743.2	191	**3.89**	575.7	94	**6.12**
35–39 years	904.3	227	**3.98**	567.0	129	**4.40**
40–44 years	1689.5	336	**5.03**	1053.9	196	**5.38**
45–49 years	3274.2	485	**6.75**	1117.1	257	**4.35**
50–54 years	4517.8	566	**7.98**	2039.9	308	**6.62**
55–59 years	5450.6	698	**7.81**	3096.9	373	**8.30**
60–64 years	6675.9	832	**8.02**	3981.9	450	**8.85**
65–69 years	7762.8	954	**8.14**	4471.0	511	**8.75**
70–74 years	7181.8	1027	**6.99**	4919.0	593	**8.30**
75–79 years	5775.4	943	**6.12**	4694.5	683	**6.87**
80–84 years	3025.3	619	**4.89**	2296.9	404	**5.69**
85–89 years	567.1	147	**3.86**	495.2	121	**4.09**
90 years and older	51.9	15	**3.46**	25.7	8	**3.21**
**Total**	48,010.1	7258	**6.61**	29,821.2	4293	**6.95**

**Table 3 jcm-12-00051-t003:** Years of potential life lost in men and women according to age of onset of dialysis or renal transplantation.

	MEN						WOMEN					
	Dialysis			Renal Transplantation			Dialysis			Renal Transplantation		
Age Onset RRT	Total YPLL	N Patient	YPLL/ Patient	Total YPLL	N Patient	YPLL/ Patient	Total YPLL	N Patient	YPLL/ Patient	Total YPLL	N Patient	YPLL/ Patient
15–29 years	325.7	52	**6.3**	64.5	166	**0.4**	423.7	37	**11.5**	62.8	129	**0.5**
30–34 years	696.6	54	**12.9**	46.6	137	**0.3**	470.8	24	**19.6**	104.9	70	**1.5**
35–39 years	657.6	52	**12.6**	246.7	175	**1.4**	423.4	33	**12.8**	143.6	96	**1.5**
40–44 years	1245.0	113	**11.0**	444.5	223	**2.0**	844.9	55	**15.4**	209.0	141	**1.5**
45–49 years	2766.9	177	**15.6**	507.3	308	**1.6**	966.7	80	**12.1**	150.4	177	**0.8**
50–54 years	3898.0	263	**14.8**	619.8	303	**2.0**	1644.4	118	**13.9**	395.5	190	**2.1**
55–59 years	4425.1	360	**12.3**	1025.5	338	**3.0**	2734.7	200	**13.7**	362.2	173	**2.1**
60–64 years	5814.7	519	**11.2**	861.2	313	**2.8**	3240.5	278	**11.7**	741.4	172	**4.3**
65–69 years	6713.2	680	**9.9**	1049.6	274	**3.8**	3936.7	370	**10.6**	534.3	141	**3.8**
70–74 years	6799.1	888	**7.7**	382.7	139	**2.8**	4632.3	523	**8.9**	286.7	70	**4.1**
75–79 years	5743.2	916	**6.3**	32.2	27	**1.2**	4659.9	671	**6.9**	34.6	12	**2.9**
80–84 years	3025.3	617	**4.9**	0.0	2	**0.0**	2296.9	404	**5.7**	0.0	0	**0.0**
85–89 years	567.1	147	**3.9**	0.0	0	**0.0**	495.2	121	**4.1**	0.0	0	**0.0**
90 years and older	51.9	15	**3.5**	0.0	0	**0.0**	25.7	8	**3.2**	0.0	0	**0.0**
**Total**	42,794.0	4853	**8.8**	5280.6	2405	**2.2**	26,795.7	2922	**9.2**	3025.5	1371	**2.2**

YPLL, years of potential life lost; RRT, renal replacement therapy.

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
