# Peer review of "Years of Potential Life Lost on Renal Replacement Therapy: Retrospective Study Cohort"

_jcm, 2022, doi:10.3390/jcm12010051_

Round 1
Reviewer 1 Report
The authors calculated the years of potential life lost on renal replacement therapy in patients receiving RRT. The strength was the large sample size and considered the influences of RT and sex. The main flaw was the Estimated years of life potentially lost have been fully attributed to RRT for all causes of death. The causes of death should be divided into RRT related or not. The patients also could be divided by their types of kidney diseases such as DN, PKD, or glomerulonephritis. Moreover, there are several problems needed to be solved.
1 The inclusion and Exclusion Criteria were not clear enough. What is the range of pediatric patients? What about cancer, CVD, PKD, or infectious diseases?
2 What was the method of calculating life expectancy?
3 Did the authors consider the role of RRT types on YPLL?
Author Response
7th December 2022
Preventive Medicine and
Public Health Department
Faculty of Medicine Faculty
University of Seville
Response to reviewers.
Dear reviewer,
Thank you very much for your interest in our work and we greatly appreciate the comments
you made to improve it. We believe that our work has improved substantially thanks to
your review.
We will now comment point by point on the revisions that the reviewer has so kindly made.
REVIEWER 1
1. “The main flaw was the Estimated years of life potentially lost have been fully
attributed to RRT for all causes of death. Causes of death should be divided into RRT
related or not. '
Response:
We completely agree with the reviewer that we overestimate premature mortality,
including all causes of death, despite only RRT-related mortality. We know of this
limitation, and it is included in the manuscript (lines 193-196, version 1).
In Western countries, mortality directly related to renal replacement therapy (RRT) is very
low. They are the cases of congestive heart failure associated with circulatory overload,
hyponatremia without urgent HD, or infection-related kidney transplantation. However,
patients with chronic kidney disease (CKD) in treatment with RRT patients have, associated
higher mortality for non-RRT related causes; higher mortality than the general population
mortality of cardiovascular etiology (1); higher mortality due infectious causes (due to
immunosuppressive treatment in transplant recipients and associated with vascular and
peritoneal catheters in dialysis patients and also higher neoplastic mortality rates because
a higher incidence of neoplasms is observed than in the general population (1).
1. Boenink R, Stel VS, Waldum-Grevbo BE, et al. Data from the ERA-EDTA Registry were
examined for trends in excess mortality in European adults on kidney replacement
therapy. Kidney Int 2020; 98: 999–1008.
We extended the limitation of our manuscript (lines 587-594 version 2) explained the
limitation in detail.
This limitation did not confer bias in the comparison between hemodialysis and kidney
transplantation because the overestimation of mortality is included in both groups and the
entire population of our study is end-stage patient kidney disease (ESKD) treated with RRT.
This higher mortality risk of patients with RRT can be considered associated with CKD, but it
is indistinguishable from that that would occur in this population if they did not require RRT,
so that this division, in this case, is impossible to complete exhaustively. Other series use the
same criteria [6, 24, 25,27,28], and this is why the mortality rate did not differ and is
comparable.
2. 'Patients could also be divided by their types of kidney disease, such as DN, PKD, or
glomerulonephritis'
Response:
The authors thank the reviewer for his suggestion. Due to the excessive length of the article,
it does not allow us to include new tables and figures. We have included recorded kidney
disease, following the EDTA-ERA criteria (1) that caused RRT in Table S1. We have made
reference to that table in the manuscript, following your suggestion; thanks again.
1. Venkat-Raman G, Tomson CRV, Gao Y, et al. New primary renal diagnosis codes
for the ERA-EDTA. Nephrol Dial Transplant 2012; 27: 4414–4419.
3. The inclusion and Exclusion Criteria were not clear enough. What is the range of
pediatric patients? What about cancer, CVD, PKD, or infectious diseases?
Response:
Pediatric patient
We excluded pediatric patients defined as patients younger than 15 years by EDTA-ERA
(2). We included patients aged 15 years and older. We corrected an error in Table 1
where the minimum age is not 16 years, but 15 years, sorry for the error.
2. Bonthuis M, Vidal E, Bjerre A, et al. Ten-year trends in epidemiology and outcomes
of pediatric kidney replacement therapy in Europe: data from the ESPN/ERA-EDTA
Registry. Pediatr Nephrol 2021; 36: 2337–2348.
CVD, cancer, or infectious disease?
In the study, we included all patients initiating RRT, including patients with cancer, CV
disease, or infectious disease diagnosis. These data are shown grouped in the article under
the Charslon comorbidity index score, which is collected at the time of inclusion of the
patient (onset of RRT). We have included a new table (Table S2) in the supplementary
material detailing the comorbidity of the patients. Furthermore, 139 patients with VHB
1.2% IC95%(1.0-1.4) and 371 patients with VHC 3.2% IC95%( (2.9-3.5) were included.
PKD
We understand that the reviewer refers to polycystic kidney disease with PKD. Patients
with PKD are also included in the study. PKD is the most prevalent disease within the group
of congenital-familial renal diseases. This disease represents 80-85% of familial
congenital kidney diseases in non-pediatric patients.
4. What was the method of calculating life expectancy?
Response:
Life expectancy estimates were obtained from life tables, abbreviated as age groups of
five years. For their calculation, they used the deaths that occurred in the calendar year
and the resident population figures at mid-year. These data are calculated and provided
by the Institute of Statistics and Cartography of Andalusia (IECA).
3 Did the authors consider the role of RRT types on YPLL?
Response:
The mean number of potential years of life lost in patients who only received dialysis was
9.0 years in both sexes, 8.8 years in men and 9.2 years in women, while among kidney
transplant recipients, this figure decreased to 2.2 years in both men and women. In other
words, the performance of a renal transplant has achieved a 75.6% reduction in the
number of YPLL in both sexes, 75.0% in men and 76.1% in women.
We hope that with all the responses to your comments, we will have the possibility that
our work will be considered for publication in your journal.
Your Sincerely
Emilio García Cabrera
Reviewer 2 Report
I have read a paper targeting the assessment of the potential years of life lost by patients on renal replacement therapy.I have various comments:
- Abbreviations should be explained in the abstract section
- Lines 48-50 – there are some missing ideas
- Line 50 – “objective indicators to indicate..” – please rewrite
- Please present the exact study period (from 01 January, to….)
- Provide the number of Ethical Committee Approval
- Line 80 – what do you mean by “morality data”?
- Line 103 – “11,5551” please correct
- Line 106 – what is “IC” (“IC95% (19.8; 21.9). The mean of the comorbidity index was 6 points IC95% (5.9;6) ranging”)
- According to your criteria, pediatric patients were excluded. But in table 1, the min age is 16 years, in Table 3 you mentioned 15 years. Please explain what pediatric patients mean to you, and why there are two different ages presented. (15 or 16 was the min age of included patients?
- Line 110 – this line is difficult to understand. Were there any patients with ESKD receiving RT without HD/PD?
- Table 1 – I do not understand what you mean by “age at death”
- Please try to write the numbers correctly during the entire manuscript (e.g. line 117 – 77,831.2, then line 118 – 29821.2, and so on)
- Line 127 – please delete the text
- All Figures and Tables should have a footer containing abbreviations and explanations, irrespective of the main text
- Line 157 – who is BDG group?
- Line 174 – 175 – “Based on our results, in older patients, the decision to start a dialysis program must be considered in the same way as in patients of productive age” – is this not the case in daily clinical practice? I think it is an exaggeration to consider this a study that is not even published. Perhaps you might reconsider these ideas.
- Lines 168 – 180 – this paragraph should be reconsidered - some aspects are biased.
- Discussion section is scarce. It should not be centered on the obtained results, but on presenting them in comparison with previous studies.
- Line 186 – explain TRS
- Lines 200-201 – please rewrite “while having received a kidney transplant, while reducing this figure by 75.6%
Author Response
7th December 2022
Preventive Medicine and
Public Health Department
Faculty of Medicine Faculty
University of Seville
Dear reviewer,
The authors are very grateful to the reviewer who pointed out formatting and typographical errors that the authors did not notice after multiple readings. The authors are deeply grateful for his improvements, which have helped to improve the work presented in the previous version.
.
We will now comment point by point on the revisions that the reviewer has so kindly made.
REVIEWER 2
We comment on all the reviewers' indications point by point.
- Abbreviations should be explained in the abstract section.
Response:
We have included in the abstract the explanations of all the abbreviations included, and we apologize for the confusion caused.
- Lines 48-50 – there are some missing ideas
Response:
Thanks again for the comment, we rewrite as
Renal transplantation is currently the best RRT option that can be applied to patients with advanced ESRD, due to lower mortality [3,4] and better patient survival [5–11] as well as better cost-effectiveness [12,13]. Access to kidney transplantation is limited primarily by organ availability. According to the latest recommendations [14], with the same clinical criteria (ESKD), age, and comorbidity being the determining factors in the choice of organ recipient. For this reason, an increasing number of renal transplants are performed each year on elderly patients [15-16]. However, there is no fixed age pattern nor objective indicators to measure survival loss between the two RRT options at different age of onset.
- Line 50 – “objective indicators to indicate..” – please rewrite
Response:
Thank you very much for your appreciation. We rewrite as follows: However, there is no fixed age pattern nor objective indicators to measure survival loss between the two RRT options.
- Please present the exact study period (from 01 January, to….)
Response:
We include the exact dates; thank you again, now the information is accurate.
- Provide the number of Ethical Committee Approval
Response:
We include the date of approval of the ethics committee in the methods section. We cannot include the code numbers because it is a digital signature and it is a long number. The ethics committee approval document is attached at the end of this response for your review and approval.
- Line 80 – what do you mean by “morality data”?
Response:
We have reviewed the entire document and have found no "morality data". Could you please indicate the exact phrase where it is located to correct it, thank you.
- Line 103 – “11,5551” please correct
Response:
We have corrected the figure, sorry again for the error.
- Line 106 – what is “IC” (“IC95% (19.8; 21.9). The mean of the comorbidity index was 6 points IC95% (5.9;6) ranging”)
Response:
We corrected all misspelled 95% confidence interval (95% CI) abbreviations. We have also included the abbreviations in the Methods section for clarity. The authors apologize to the reviewer for all these errors due to translation.
- According to your criteria, pediatric patients were excluded. But in table 1, the min age is 16 years, in Table 3 you mentioned 15 years. Please explain what pediatric patients mean to you, and why there are two different ages presented. (15 or 16 was the min age of included patients?
Response:
As we comment on Review 1, we excluded pediatric patients defined as patients younger than 15 years of age by EDTA-ERA (1). We included patients aged 15 years and older. We corrected an error in Table 1 where the minimum age is not 16 years, but 15 years, sorry for the error.
- Bonthuis M, Vidal E, Bjerre A, et al. Ten-year trends in epidemiology and outcomes of pediatric kidney replacement therapy in Europe: data from the ESPN/ERA-EDTA Registry. Pediatr Nephrol 2021; 36: 2337–2348.
- Line 110 – this line is difficult to understand. Were there any patients with ESKD receiving RT without HD/PD?
Response:
We changed the line for a total of 3776 patients, 32.7% of the entire ESKD patient studied cohort 95% CI (31.8; 33.5) receiving RT. For better understanding.
- Table 1 – I do not understand what you mean by “age at death”
Response:
‘Age at death’ means the age at the time the patient died. If the reviewer considers that this is not an appropriate or clear enough term, please indicate it, and we will change it. Thank you.
- Please try to write the numbers correctly during the entire manuscript (e.g. line 117 – 77,831.2, then line 118 – 29821.2, and so on)
Response:
We have rewritten all the numbers included in the tables by removing the commas from the units of thousands, and we are sorry for the confusion caused.
- Line 127 – please delete the text
Response:
We maintain the text, in the next section heading, but we have changed format. It is better now, thank you.
- All figures and tables should have a footer containing abbreviations and explanations, regardless of the main
Response:
We have included the meaning of the abbreviations in the figures and tables; thank you for the comment, now it is clearer.
- Line 157 – who is BDG group?--> Grupo del Global Burden Disease GBD) Study
Response:
We have included the meaning of the abbreviations; thank you again for the comment.
- Line 174 – 175 – “Based on our results, in older patients, the decision to start a dialysis program must be considered in the same way as in patients of productive age” – is this not the case in daily clinical practice? I think it is an exaggeration to consider this a study that is not even published. Perhaps you might reconsider these ideas.
- Lines 168 – 180 – this paragraph should be reconsidered - some aspects are biased.
- Discussion section is scarce. It should not be centered on the obtained results, but on presenting them in comparison with previous studies.
Responses 16 to 18:
We are terribly sorry for all the errors that have occurred in the discussion of the results. We have changed the structure and content of it almost completely.
- Line 186 – explain TRS
Response:
It was a translation error, it was really RRT.
- Lines 200-201 – please rewrite “while having received a kidney transplant, while reducing this figure by 75.6%.
Response:
We have rewritten this sentence.
We hope that with all the responses to your comments, we will have the possibility that our work will be considered for publication in your journal.
Your Sincerely
Emilio García Cabrera

Round 2
Reviewer 1 Report
None.
Author Response
14th December 2022
Preventive Medicine and
Public Health Department
Faculty of Medicine Faculty
University of Seville
The manuscript reads better in light of the revisions which have been undertaken. However, there are still some items which need to be addressed.
Dear Academic Editor,
I am writing on behalf of all authors to express our sincere gratitude for taking the time to review our manuscript titled Years of potential live lost on renal replacement therapy. Re-trospective study cohort. Your feedback and insights have been invaluable in helping us improve the quality of our work.
We will now comment point by point on the revisions that you have so kindly made.
- The Figures appear redundant in light of the summary information which is provided in the Tables-ie the information in Table 2 is similar to that in Figure 1. The same applies to the information in Table 3 being similar to that depicted in Figure 2. Plus, the quality of the Figures is such that they are difficult to visualize. Can the authors please address this.
Response: We did remove Figure 2 following the suggestion of the editor because the information added to Table 3 is not significant. We improve the resolution of figure 1 and maintain it, because it represents the YPLL in every year during the study period and this information is not at table.
- The dialysis and transplant cohorts are not the same, hence this limits the validity of the summary YPPL data-noting that no patients aged over 80 underwent renal transplantation. So, there will be a significant selection bias underpinning these two cohorts to begin with -which needs to be more adequately covered off in the limitations of the study in the discussion section, along with also mentioning the time lag bias (ie not all patients entered into the study were exposed to the same time period duration of follow up).
Response:
We deeply appreciate the comment of the editor about octogenarian patients. There is an increase experience in this area, but it is still limited because there are not many patients who are candidates for transplantation over 80 years, mainly due to the associated comorbidity. We added this limitation of our results in the discussion
Third, only 2 patients over 80 years of age in our cohort, receiving renal transplantation, overestimate the real difference between HD and RT, however recent published data [30] show better survival rates over RT patients, in line with our results.
- Why is there a difference in the YPPL between the men and the women? Can you provide some explanation in the discussion section. What are the possible factors here?
Response: Thank you again for your comments. We added an entire paragraph at the discussion about the different mortality rate by gender.
Females receiving HD treatment do not have a significant survival advantage over compared to the general population [24]. The prognostic factors associated with the female gender that cause this decrease in survival remain controversial. A less protective effect of body mass index (BMI) on HD mortality, associated with a different rate of skeletal muscle and fat due to sex hormone is one of the most recent hypotesis [25]. However, the effect of diabetes is the most reported interaction factor on the different survival rates observed between sexes in all causes of death [24-26].
- Finally, the conclusions need to be strengthened as to where the greatest benefit with respect to mitigating YPLL might be able to be gained. This appears to be in the 30-64 year age groups (mention of this also needs to be made in the discussion section of the manuscript as well).
Response: After remodeling the discussion to incorporate gender results, we have added a small paragraph to the conclusion. We again thank you for your work because we think the conclusions have been much stronger after your review.
Since the limiting factor of organ availability cannot be ignored, greater monitoring or specific programs for elderly people, especially diabetic women, must be implemented to mitigate the impact of premature death due to dialysis.
I believe these changes have improved the overall quality of the manuscript and hope that you will find it suitable for publication. I appreciate the time and effort you have taken to review our work, and We are grateful for any further feedback you may have.
Your Sincerely
Emilio García Cabrera